# DORN1 Is Involved in Drought Stress Tolerance through a Ca^2+^-Dependent Pathway

**DOI:** 10.3390/ijms232214213

**Published:** 2022-11-17

**Authors:** Qingwen Wang, Hongbao Bai, Ahmad Zada, Qingsong Jiao

**Affiliations:** 1College of Bioscience and Biotechnology, Yangzhou University, Yangzhou 225009, China; 2National Key Laboratory of Crop Genetics and Germplasm Enhancement, College of Life Sciences, Nanjing Agricultural University, Nanjing 210046, China; 3Joint International Research Laboratory of Agriculture and Agri-Product Safety of the Ministry of Education of China, Yangzhou University, Yangzhou 225009, China

**Keywords:** Ca^2+^, chlorophyll fluorescence, DORN1, drought stress, stomata, water loss

## Abstract

Water shortages caused by climate change seriously threaten the survival and production of plants and are also one of the major environmental pressures faced by plants. DORN1 was the first identified purinoceptor for the plant response to extracellular ATP. It has been established that DORN1 could play key roles in a series of biological activities in plants. However, the biological roles of DORN1 and the mechanism remain unclear under drought stress conditions in plants. Here, DORN1 was targeted for knockout by using the CRISPR/Cas 9 system. It was found that the loss function of DORN1 resulted in a significant decrease in the effective quantum yield of PSII [Y(II)], the photochemical quenching coefficient (qP), and the rate of photosynthetic electron transport through PSII (ETR), which reflected plants’ photochemical efficiency. Whereas Y_(NO)_ values showed obvious enhancement under drought stress conditions. Further experimental results showed that the Y_(II)_, q_P_, and ETR, which reflect plants’ photochemical efficiency, increased significantly with CaCl_2_ treatment. These results indicated that the drought tolerance of the mutant was decreased, and the exogenous application of calcium ions could effectively promote the drought tolerance of the *dorn1* mutant. Transpiration loss controlled by stomata is closely related to drought tolerance, further, we examined the transpirational water loss in *dorn1* and found that it was greater than wild-type (WT). Besides, the *dorn1* mutant’s stomatal aperture significantly increased compared with the WT and the stomata of *dorn1* mutant plants tend to close after CaCl_2_ treatment. Taken together, our results show that DORN1 plays a key role in drought stress tolerance in plants, which may depend on calcium and calcium-related signaling pathways.

## 1. Introduction

Drought stress is one of the biggest stresses on plant growth and development. Recently, it has become more accentuated due to global warming. Stomata are leaf pores that control gas exchange and water transport, and therefore, impact critical functions such as photosynthesis and drought tolerance. Studies have shown that Ca^2+^-dependent signaling plays a role in regulating stomatal movement [1], and this Ca^2+^-regulated stomatal movement is crucial to water use efficiency and drought tolerance in plants [2,3].

Besides being a cellular energy source, the adenosine triphosphate (ATP) can be transferred to the extracellular space through wound leakage, secretion pathways, or membrane transporter. Extracellular ATP (eATP) could act as a famous signal molecule used by plants and functions in various processes, such as growth, development, and stress responses [4,5,6]. These functions depend on the cell surface receptor eATP receptors and activate second messenger molecules intracellularly, such as cytosolic free Ca^2+^ ([Ca^2+^]_cyt_) and hydrogen peroxide [7,8]. As one of the famous second messengers, [Ca^2+^]_cyt_ plays a vital role in plants’ development and stress tolerance [3,9,10]. Such as brassinosteroid biosynthesis and plant growth [11], balancing plant growth and immunity [12], and enhancing normal pollen tube growth and fertilization [13]. Recently, does not respond to nucleotides (DORN1, also known as P2K1) and has been identified as the first purinoceptor on the surface of the plant cell, which is essential for plant response to ATP [14]. DORN1 is a legume-type lectin receptor kinase that became widely known as an eATP receptor through its role in increasing [Ca^2+^]_cyt_ in seedlings [14]. Transgenic plants overexpressing DORN1 showed longer roots length, a greater rosette surface area, and more resistance to the pathogen [15,16]. The *dorn1* mutant plants had a reduced transcriptional response to both ATP treatment and wounding [17], as well as pathogen resistance [18]. Although there is mounting evidence that DORN1 plays a role in plant biotic stress tolerance, the mechanisms of signal transduction through DORN1 remain largely unknown.

eATP and its receptor DORN1 have been established as an important signaling pathway responding to salt stress tolerance [19], chilling response [20], and biotic stress resistance [21,22] in plants. Drought stress is a major factor limiting plant growth and development in nature, and it is also an important factor affecting crop production. However, little is known about whether DORN1 is involved in drought stress tolerance and the possible mechanisms. As a rapid, noninvasive technique for detecting subtle differences in leaf metabolism or developmental disorders of the seedlings [23]. It has long been known that the chlorophyll fluorescence induction kinetics of leaves from plants provide an indicator of plant physiological state and are sensitive to changes in stress responses [24]. The activity of photosystem II (PSII) declines more rapidly than any other physiological activity [25,26]. In the present study, we created the *dorn1* mutant by CRISPR/Cas9 and found that the loss functions of DORN1 impaired drought stress tolerance in plants, which may be related to the Ca^2+^-regulated stomatal movement. These findings revealed the role and underlying mechanism of DORN1 in maintaining drought tolerance in plants and also offered a strategy for enhancing plant drought tolerance.

## 2. Results

### 2.1. Creation and Identification of the Loss Function Mutants for DORN1

CRISPR/Cas9 provided a convenient tool for creating loss-of-function mutants for genes of interest. To create mutations in DORN1, synthetic guide RNA targeting the first exon of DORN1 was designed by the E-CRISP tool. Finally, we successfully achieved site-specific deletion of DORN1, *dorn1-c3* with an 8 bp deletion, and *dorn1-c6* with a 7 bp insertion and an 8 bp deletion (Figure 1A,B). Phenotypic analyses showed the mutant plants could grow normally under normal conditions, and there was no significant difference in fresh weight analysis (Figure 1C,D).

### 2.2. The Survival Rate Was Decreased at Day 10 of Soil drying in the dorn1 Mutant

Water shortages are a global environmental threat that seriously threatens plants. To determine whether the DORN1 functions in drought tolerance, we further confirmed this by phenotypic analysis of plants with direct drought treatment. After 10 days of drought treatment, the *dorn1* mutant plants mutant showed obvious severe dehydration and could not survive after rehydration (Figure 2A). The survival rate is often taken as an indicator of the drought tolerance of plants. WT and the *dorn1* mutant plants were rehydrated after 10 days of drought treatment, and the survival rate was calculated by counting the number of surviving plants at 2 days after rewatering the plants. The ratio of surviving plants to total treated plants was then calculated. Statistical analysis showed that the survival rate of the mutant decreased significantly after rehydration (Figure 2B).

### 2.3. Loss Function of DORN1 Has No Obvious Influence on F_v_/F_m_ Performance

The maximum yield of the PSII photochemical reactions (F_v_/F_m_) reflects the maximum quantum efficiency of PSII photochemistry, and it has been widely used for early stress detection in plants [27]. Using a chlorophyll fluorometer, we first examined whether this mutation could affect the F_v_/F_m_. The results showed that there were no reliable changes in F_v_/F_m_ under normal conditions. While F_v_/F_m_ was characterized by a slight downward trend compared to the WT under drought stress conditions, the difference did not reach statistical significance (Figure 3A). Similarly, there were no significant changes in F_v_/F_m_ after treatment with CaCl_2,_ neither in the *dorn1* mutant nor the WT (Figure 3A).

### 2.4. Exogenous CaCl_2_ Promotes the Photochemical Reaction Efficiency Performance in dorn1 Plants under Drought Stress

To further investigate the function of DORN1 in the actual photochemical efficiency of plants, we examined the major photosynthetic index under normal and drought conditions. Y_(II)_ denotes the effective quantum yield of PSII, which is an indicator of the actual photochemical activity of PSII [28]. The results showed that no difference in the Y_(II)_ of the *dorn1* mutant compared with the WT under normal growth conditions. It is worth noting that Y_(II)_ was significantly reduced in the *dorn1* mutant under drought conditions, although no visible changes were observed in the WT under drought conditions. However, after treatment with CaCl_2_, Y_(II)_ of the *dorn1* mutant increased significantly and returned to WT levels (Figure 3B). The photochemical quenching coefficient (q_P_) indicates a fraction of “open” PSII reaction centers based on a puddle model [29,30]. In the *dorn1* mutant, q_P_ was significantly reduced when compared with WT under drought conditions, but this decrease was effectively delayed or even improved to the WT levels by the exogenous CaCl_2_ treatment (Figure 3C).

The actual photochemical efficiency of PSII is considered a reasonable index of the quantum yield of photosynthetic electron transport. To perform these analyses Next, we examined the rate of photosynthetic electron transport directly through PSII (ETR). The results showed that ETR had the same changing tendency as the Y_(II)_ and q_P_. Drought treatments resulted in a significant reduction of ETR levels in the *dorn1* mutant, and CaCl_2_ treatment significantly recovered ETR levels in the *dorn1* mutant under drought stress conditions. whereas there was no obvious effect on the ETR levels of WT plants under similar treatment (Figure 3D).

### 2.5. Photooxidative Damage Increased in the dorn1 Plant under Drought Stress

The quantum yield of non-light-induced non-photochemical fluorescence quenching Y_(NO)_ (also denoted as Φ_f.D_) reflects the yield for other energy losses [29]. It is generally accepted that high Y_(NO)_ indicates both photochemical energy conversion and protective regulatory mechanisms are inefficient. Plants may have serious problems coping with the incident radiation; either they had been damaged already or they would be photodamaged upon further irradiation [31]. Results showed that there was no visible change in Y_(NO)_ between the *dorn1* mutant and WT plants under normal growth conditions (Figure 4A). As a sensitive indicator of the plant stress response, Y_(NO)_ increased substantially due to drought stress in the *dorn1* mutant relative to the WT. Y_(NO)_ was characterized by a slight downward trend in the *dorn1* mutant by the exogenous CaCl_2_ treatment, rather than an increasing trend in the WT plants (Figure 4A).

Y_(NPQ)_ represents the quantum yield of light-induced non-photochemical fluorescence quenching, without such dissipation there would be the formation of singlet oxygen and reactive radicals, which cause irreversible damage. On the one hand, a high Y_(NPQ)_ value indicates that the photon flux density is excessive. On the other hand, it shows that the sample has retained the physiological means to protect itself by regulation. No change was observed in Y_(NPQ)_ between the *dorn1* mutant and WT plants under normal growth conditions (Figure 4B). The *dorn1* mutant showed a clear decrease in Y_(NPQ)_ relative to the WT under drought conditions, while it showed an increasing trend after treatment by the exogenous CaCl_2_ (Figure 4B). These results suggest that exogenous CaCl_2_ could alleviate photooxidative damage induced by drought stress in the *dorn1* plant.

### 2.6. The dorn1 Plants Have a Greater Transpirational Water Loss

The *dorn1* plants showed an accelerated decline in photosynthesis under drought stress conditions, indicating that drought tolerance performance in the mutant seedlings was impaired. We further confirmed this by phenotypic analysis of plants with direct drought treatment (Figure 2). It is known that transpirational water loss has a decisive effect on plant water metabolism and drought resistance in the plant. To investigate whether the reduced drought tolerance in this study may be attributed to transpirational water loss, we examined the rate of transpirational water loss rate both in WT and the *dorn1* mutant. The two mutants’ transpiration water loss rates of the showed an increasing trend compared with the WT (Figure 5).

### 2.7. Exogenous CaCl_2_ Could Relieve Excessive Stomatal Opening in the dorn1 Mutants

Since more than 90% of water loss occurs through transpiration from the stomata in terrestrial plants [32,33]. We further analyzed the stomatal phenotypes in WT and the *dorn1* mutant. As we expected, the stomatal aperture of the mutant was significantly increased compared with the WT (Figure 6A). Besides, stomatal opening showed a decreasing trend after treatment by the exogenous CaCl_2_ (Figure 6B). All these results suggested that excessive water loss caused by increased stomatal aperture is an important reason for the decreased drought tolerance of the *dorn1* plants. Furthermore, according to the experimental results and previous reports, we also proposed a working model for the roles of DORN1 in plant drought response (Figure 6C).

## 3. Discussion

The eATP has attracted more and more attention since it was found to be a signaling molecule. In 2014, Choi et al. identified the first ATP receptor-DORN1 in the model plant *Arabidopsis thaliana*. eATP is recognized by DORN1 on the cell surface and dependents on downstream signaling pathways activation. Numerous studies have shown that eATP plays multiple roles in plant development and stress response [6], such as seed germination, photosynthesis, plant immune response, and salt tolerance. It was reported that these signaling pathways are closely related to the induction of intracellular second messengers, such as [Ca^2+^]_cyt_ [5]. Drought is one of the most important and prevalent stress factors for plants. In the present study, we found that the photochemical efficiency of PSII parameters was significantly lower in *dorn1* mutants than in WT under drought conditions. These results suggested that the drought tolerance of the mutant plants was decreased. In addition, exogenous CaCl_2_ alleviates drought stress on the photosynthetic apparatus of the plants. We also observed that the *dorn1* mutants showed an increased water loss rate, which was accompanied by the opening status of leaf stomata, the main channel for water loss through transpiration. Moreover, the application of CaCl_2_ could effectively alleviate the excessive opening of stomata (Figure 6). Based on our results, it is attractive to hypothesize that DORN1 plays a vital role in drought stress tolerance in *Arabidopsis thaliana*, and this may be associated with [Ca^2+^]_cyt_.-induced stomatal closure.

Although no obvious phenotype was observed in the *dorn1* mutants compared with WT when grown under optimal growth conditions. The latest study has shown that Arabidopsis lectin receptor kinase P2K2 is a second plant receptor for extracellular ATP and contributes to innate immunity [34]. These findings suggest that there may be functional redundancy between the two eATP receptors, which could explain the normal phenotype of the *dorn1* mutants under optimal growth conditions. A recent report elucidated that DORN1 may be an important signal that helps mediate water homeostasis [35], especially under conditions of drought. Here we found that drought tolerance was significantly reduced in the *dorn1* mutants, our results confirmed the conclusion from previous research that DORN1 plays a role in water homeostasis [35]. Plants regularly face adverse growth conditions, such as drought and plant disease. Chlorophyll fluorescence is a powerful and widely used indicator of the physiological status of plants under abiotic or biotic stress [25]. We found Y_(II)_, q_P_ and ETR showed a significant reduction in the *dorn1* mutant under drought stress conditions, while exogenous CaCl_2_ could alleviate drought stress on the photosynthetic apparatus of these parameters. These results suggested that the loss function of DORN1 led to decreased photosynthetic capacity in plants, which might be related to [Ca^2+^]_cyt_. Y_(NO)_ is considered an indicator for PSII photodamage. Under normal conditions, the rate of repair of PS II is balanced with the rate of damage. Drought stress reduced the rate of repair and disturbed the dynamic balance, which led to the decreased activity of PS II. Therefore, it is also worth investigating whether overexpression of DORN1 can promote drought tolerance in plants. Furthermore, CaCl_2_ could reduce the increased Y_(NO)_ caused by drought stress in the *dorn1* mutant. This finding suggested the *dorn1* mutant exhibited more severe photo or oxidative damage under drought conditions, and this damage is also closely related to [Ca^2+^]_cyt_.

Studies have shown [Ca^2+^]_cyt_ as a specific second messenger through the plasma membrane DORN1 receptor in plants [36,37]. eATP could cause the production of reactive oxygen species by NADPH oxidase AtRBOHC/AtRBOHD [6]. This may further activate calcium channels in the plasma membrane, which contributes to the elevation of cytosolic-free Ca^2+^ [7]. This is also associated with the phenomenon of Ca^2+^ waves, in which a [Ca^2+^]_cyt_ signal could propagate along with tissues [14,38]. The Ca^2+^ level in the *dorn1* mutant was significantly lower than that in the wild type after ATP treatment [14]. Moreover, [Ca^2+^]_cyt_ is critical for stomatal movement, application of exogenous Ca^2+^ to plant tissue induced stomatal closure [39,40], external calcium elicited cytosolic calcium oscillations, and induced stomatal closure. Consistent with previous reports, our results showed that exogenous Ca^2+^ alleviated the stomatal aperture in the mutant deficient in DORN1. Calcium-permeable channels in the plasma membrane (PM) of guard cells play an important role in Ca^2+^-regulated stomatal closure [40]. Therefore, the relationship between DORN1 and calcium channel proteins deserves further discussion. Studies have shown that exogenous calcium effectively improves plant photosynthesis under heat or cold stress conditions [41,42]. Similarly, exogenous CaCl_2_ treatment significantly enhanced Y_(II)_, q_P,_ and ETR levels in the *dorn1* mutant under drought stress conditions. Besides, the increase in Y_(NO)_ in the *dorn1* mutant was also alleviated by exogenous CaCl_2_ treatment. These findings showed that Ca^2+^ is essential in restoring the rate of repair and disturbs the dynamic balance of PS II in the *dorn1* mutant drought stress condition.

We show that DORN1 regulates drought avoidance via modulating [Ca^2+^]_cyt_.-induced stomatal closure, which may provide a link between the DORN1-associated eATP signaling pathway and plant drought stress tolerance.

## 4. Materials and Methods

### 4.1. Plant Materials and Growth Condition

All Arabidopsis plants used in the current study were in the Columbia-0 background. The seeds were surface sterilized with 75% ethanol containing 0.05% Tween-20. The seeds were placed on plates that contained 1/2 MS medium (Duchefa, M0222) supplemented with 0.6% agar (Sangon Biotech, A505255) and 1% sucrose (Sangon Biotech, A502792), and then stratified at 4 °C for 2 days in darkness. The plates were then placed for seed germination and plant growth in a growth chamber set at 22 °C under 110 ± 5 μmol m^−2^ s^−1^ light of long-day conditions with a 16 h light/8 h dark cycle, then plants transferred to nutrient soil after one week of growth. For the CaCl_2_ treatment, seedlings were grown on nutrient soil for the indicated number of days, and then 1 mM CaCl_2_ was sprayed on the leaves once per day (1 mL per plant).

### 4.2. Construction of CRISPR/Cas9 Plant Expression Vectors

Nucleotide sequence encoding for DORN1 (AT5G60300) was retrieved from the Arabidopsis Information Resource (https://www.arabidopsis.org/, accessed on 20 December 2017). Synthetic guide RNA (AT5G60300-sgRNA:5′-GATATTGACCACAATCATGT-3′) targeting the first exon was designed by the E-CRISP tool (http://www.e-crisp.org, (accessed on 20 December 2017). The oligomer of designed sgRNA was synthesized and cloned into the pKSE401 binary vector carrying the Cas9 gene (Addgene plasmid # 62202).

### 4.3. Identification of the Homozygous Mutant Plants

Genomic DNA was extracted from the positive T_1_ plants’ leaves, as the template for PCR reaction. The DNA was amplified with the forward primer DORN1-F 5′-CAGTTCATCTGGACCACTCT-3′; Reverse primer DORN1-R 5′-CACTGCTATTTCTCTGCCTT-3′. The PCR-amplified fragment of each DNA sample was sequenced and compared to the gene sequence of DORN1 in WT by SnapGene software (www.snapgene.com, (accessed on 17 May 2018).

### 4.4. Measurement of Chlorophyll Fluorescence

Chlorophyll fluorescence parameters were measured using Image-PAM with a MAXI measuring head (Image–PAM, WALZ, Effeltrich, Germany), as described previously by Demmig-Adams [43]. F_v_/F_m_, the PSII maximal photochemical efficiency was defined as (F_m_ − F_0_)/F_m_, where F_m_ is the maximum fluorescence emission from the dark-adapted state measured with a pulse of saturating light and F_0_ is the minimal fluorescence from the dark-adapted state. Y_(II)_, the effective quantum yield of PSII photochemistry was defined as (F_m_’ − F_s_)/F_m_’, where F_m_’ is the maximum fluorescence emission from the light-adapted state measured with a pulse of saturating light and F_s_ is the steady-state level of fluorescence emission at 110 μmol m^−2^ s^−1^. q_P_, was defined as (F_m_’ − F_s_)/(F_m_’ − F_0_’) according to Lazár [29]. As Y_(II)_ represents the number of electrons transferred per photons absorbed by PSII, the ETR (the rate of photosynthetic electron transport through PSII can be calculated as ETR = Y_(II)_ × PAR × 0.5 × 0.84 as per Genty [44]. Y_(NO)_, the quantum yield of non-regulated energy dissipation of PSII, and Y_(NPQ)_, the quantum yield of regulated energy dissipation of PSII were also defined by Kramer [31].

### 4.5. Measurement of Transpirational Water Loss

Water loss was measured as described previously with minor modifications [45]. Shoots of four-week-old relative to the WT and *dorn1*-c3/c6 were detached from plants cultivated in the growth chamber and quickly weighed to obtain fresh weights (W_0_). The shoots were then placed in filter paper under the light intensity of 110 ± 5 μmol m^−2^ s^−1^ at 25 °C and 50% relative humidity. The weight was recorded every 30 min (W_1_, W_2_, W_3_, W_4_), and the loss of fresh weight ([W_0_ − W_1_]/W_0_, [W _0_− W_2_,]/W_0_, [W_0_ − W_n_…]/W_0_,) was used to indicate the transpirational water loss rate.

### 4.6. Measurement of Stomatal Closure in Response to CaCl_2_ and Drought Treatment

The leaves from the same position within the rosette were preserved in the wash solution (ethanol/acetic acid, 19:1) at room temperature for 3–5 h, washed with distilled water, and placed in the clear solution (glycerol/chloral hydrate/water, 1:8:1, vol/wt/vol) overnight or for a few days [46]. Four images at 400× magnification were captured per leaf using a Zeiss LSM 880 NLO system (Carl Zeiss Microscopy GmbH, Jena, Germany) with a 40× water objective and processed using Adobe Photoshop. Stomatal aperture width and length were measured from the images using the Image J software (ImageJ nih.gov).

### 4.7. Statistical Analysis

The results were expressed as the mean ± standard deviation. The data were statistically evaluated by one-way ANOVA between the different groups and Tukey’s multiple comparisons test; The difference was considered to be statistically significant when *p* < 0.05.

## Figures and Tables

**Figure 1 ijms-23-14213-f001:**
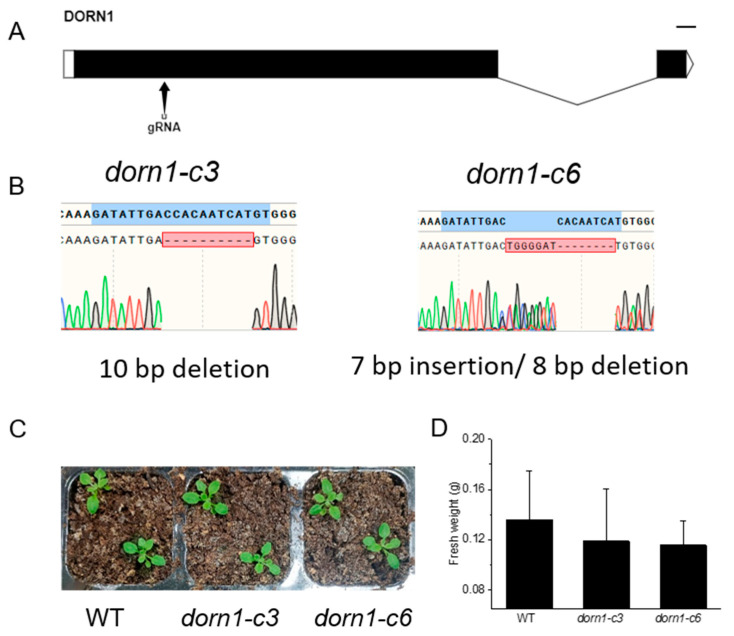
The *dorn1* mutants showed no obvious phenotype under normal growth conditions. (**A**) Gene structure of DORN1 and the Cas9/guide RNA target site. (**B**) Mutation site information of the *dorn1-c3* and *dorn1-c6*, *dorn1-c3* characterized a 10 bp deletion and *dorn1-c6* with 7 bp insertion/8 bp deletion. (**C**) The *dorn1* mutants grew well under normal culture conditions. (**D**) Statistical analysis of fresh plant weight. Each data point represents the average fresh weight of ~15 seedlings of three duplicate experiments, and error bars represent the standard deviation (SD).

**Figure 2 ijms-23-14213-f002:**
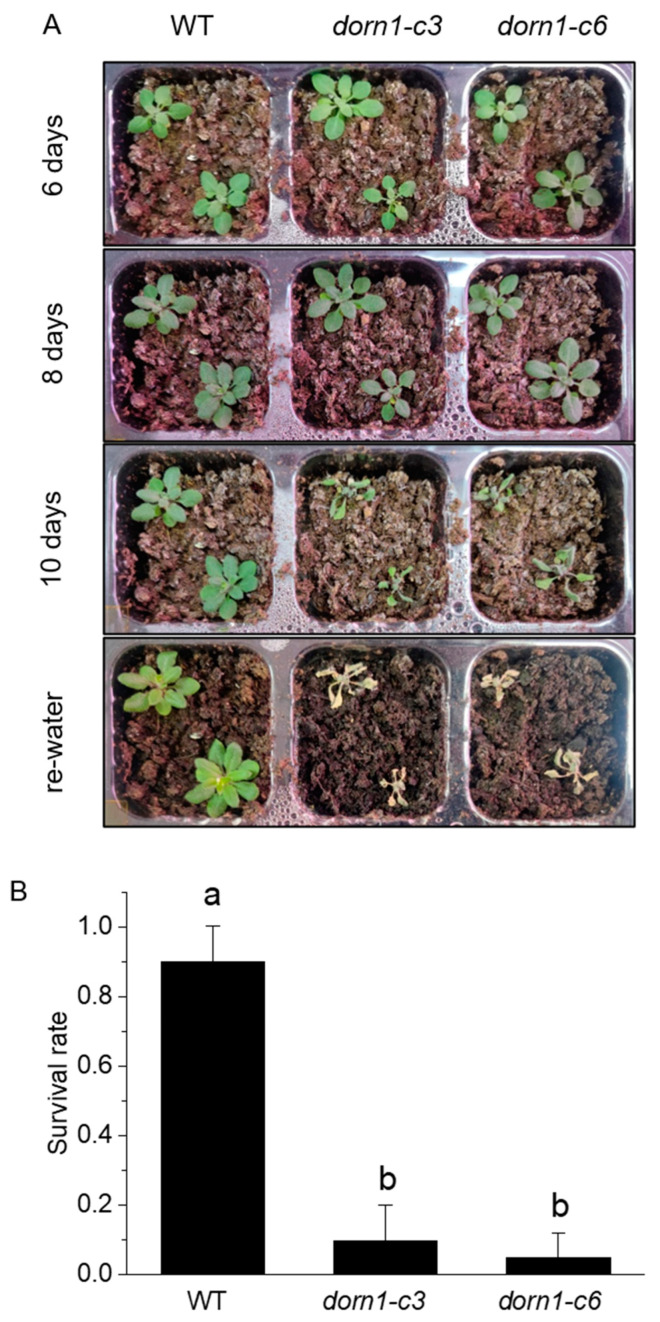
Phenotypic analysis of drought tolerance in the *dorn1* mutant. (**A**) Phenotype of 7-day seedlings treated with different periods of drought and rehydration. (**B**) The survival rate of different plants after rehydration, different letters indicate significant differences (*p* < 0.05) using ANOVA analysis. Each data point represents the average survival rate of ~15 seedlings of three duplicate experiments, and error bars represent the standard deviation (SD).

**Figure 3 ijms-23-14213-f003:**
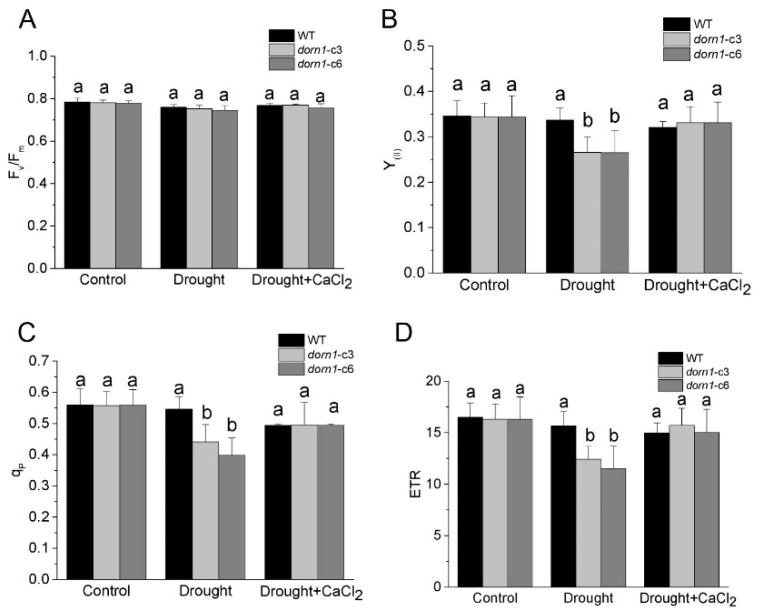
Effects of CaCl_2_ on the photochemical reaction performance of the *dorn1* plant under drought stress. Effects of CaCl_2_ on the performance of F_v_/F_m_ (**A**), Y_(II)_ (**B**), q_P_(**C**) and ETR (**D**) in the *dorn1* mutant and WT plants. The treatment group was pre-treated by spraying 1 mM CaCl_2_ once a day for three days. Different letters indicate significant differences (*p* < 0.05) using ANOVA analysis. Each data point represents the average data of ~15 seedlings of three duplicate experiments, and error bars represent the standard deviation (SD).

**Figure 4 ijms-23-14213-f004:**
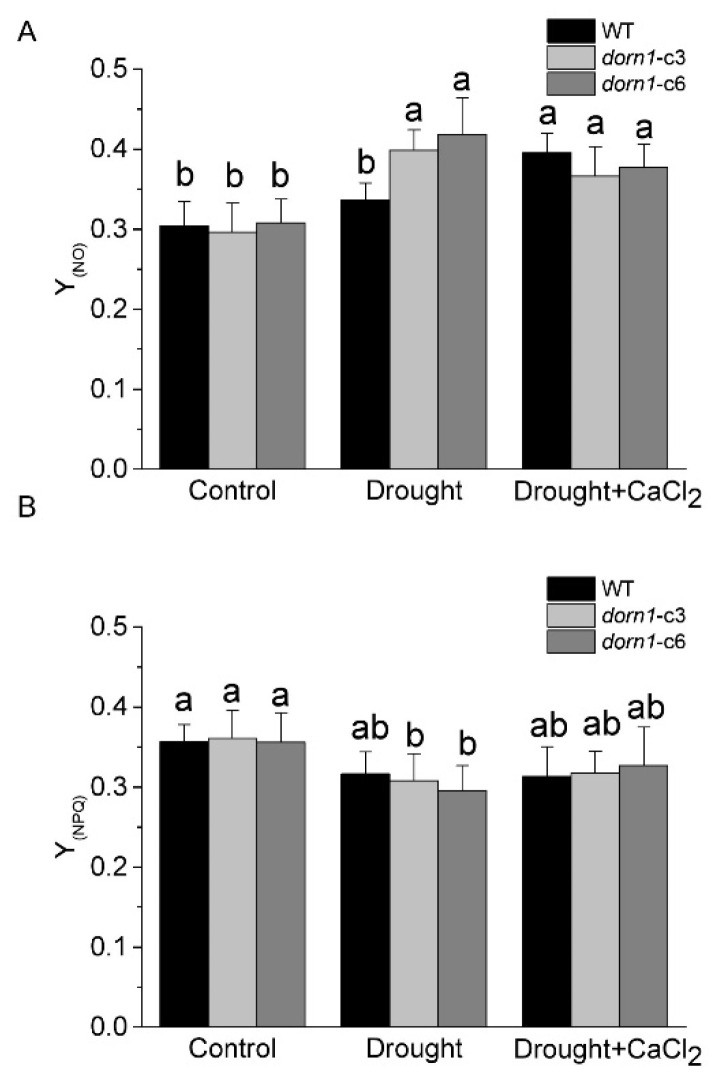
Effects of CaCl_2_ on photooxidative damage of the *dorn1* plant under drought stress. Effects of CaCl_2_ on Y_(NO)_ (**A**) and Y_(NPQ)_ (**B**) performance in the *dorn1* mutant and WT plants. The treatment group was pre-treated by spraying 1 mM CaCl_2_ once a day for three days. Different letters indicate significant differences (*p* < 0.05) using ANOVA analysis. Each data point represents the average data of ~15 seedlings of three duplicate experiments, and error bars represent the standard deviation (SD).

**Figure 5 ijms-23-14213-f005:**
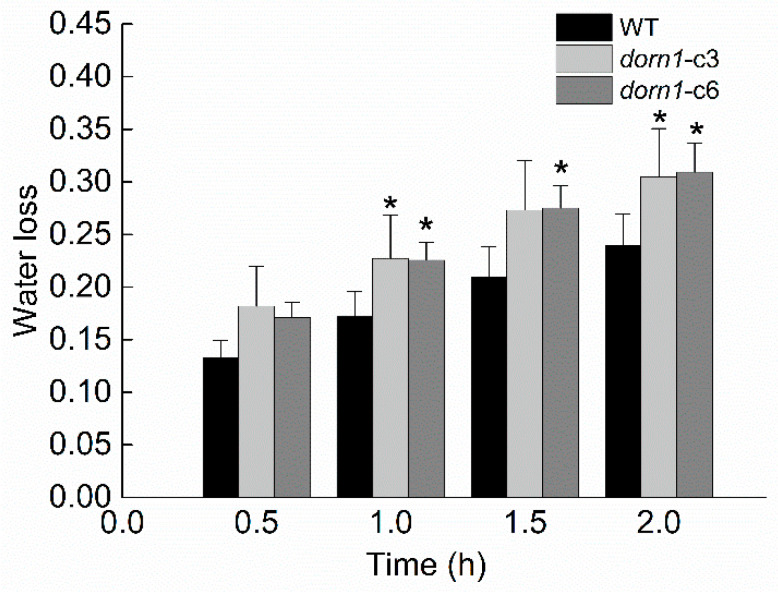
The transpirational water loss of the *dorn1* and WT plants. Mutant and WT plants are grown on soil for 21 days, then the water loss rate is measured and analyzed. Each data point represents the average water loss of ~15 seedlings of three duplicate experiments, and error bars represent the standard deviation (SD). “*“ indicate significant differences (*p* < 0.05) using Tukey’s analysis.

**Figure 6 ijms-23-14213-f006:**
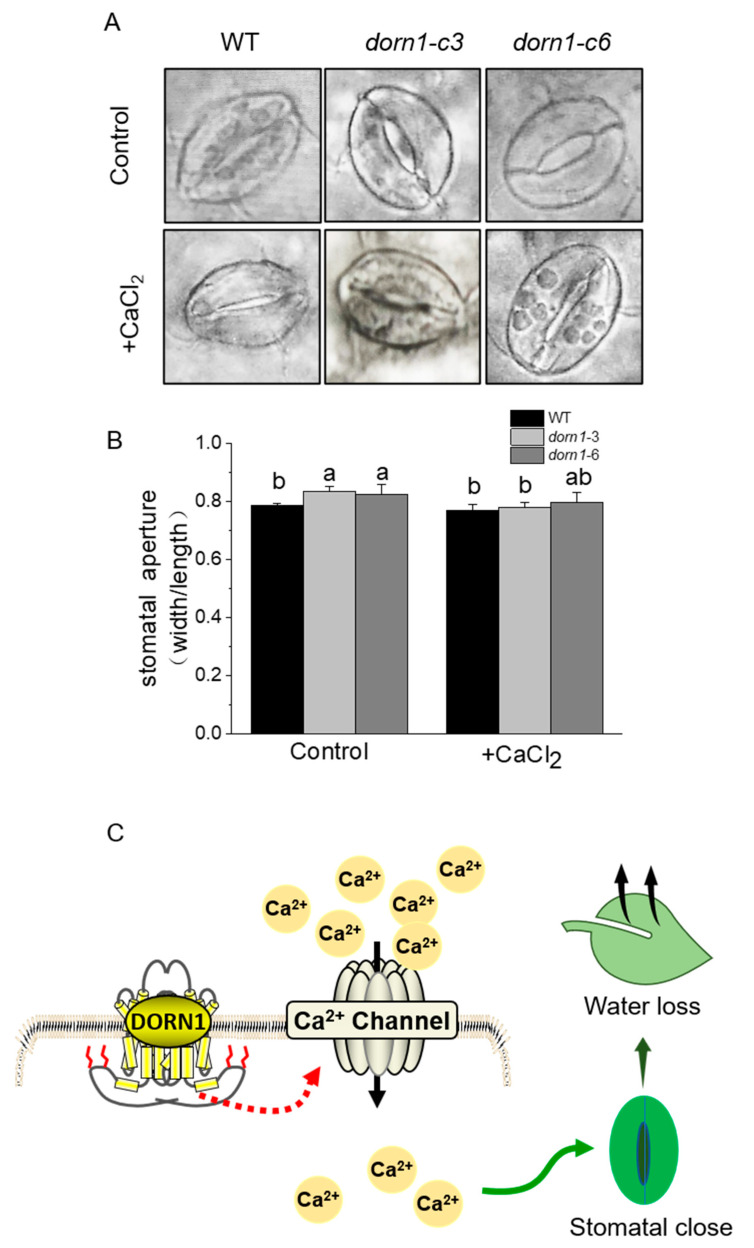
Effects of CaCl_2_ on stomatal aperture. (**A**) Stomatal phenotype analysis of the plants with or without 1 mM CaCl_2_ treatment, for the treatment group, plants were pre-treated by spraying 1 mM CaCl_2_ one hour before decoloring. (**B**) Statistical analysis of the stomatal aperture (width/length). Different letters indicate significant differences (*p* < 0.05) using ANOVA analysis. Each data point represents the average stomatal aperture of ~15 seedlings of three duplicate experiments, and error bars represent standard deviation (SD). (**C**) A possible working model for DORN1 in drought tolerance in plants.

## Data Availability

Not applicable.

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
