# Peer review of "DORN1 Is Involved in Drought Stress Tolerance through a Ca^2+^-Dependent Pathway"

_ijms, 2022, doi:10.3390/ijms232214213_

Round 1
Reviewer 1 Report
In the manuscript entitled DORN1 is involved in drought stress tolerance through a Ca2+-dependent pathway, Wang et al have characterized the function of the DORN1 gene from Arabidopsis. Through various physiological and phenotypic analyses, the authors have concluded that DORN1 functions in drought tolerance. Compared to the wild type, the CRISPR/Cas9 edited plants of DORN1 showed decreased performance under drought conditions which was observed by increased water loss, stomatal opening, and reduced survival rate in the mutant plants. Overall, the manuscript is well presented with the required results supporting the conclusion. Comments: Line 79: Change to "exon" Line 98: Should be "dorn1 mutant plants" Result 2.1: Only keep the results here. Delete the methodology part Result 2.2 Describe how the survival rate was calculated. Figure 5: Statistical analysis should be performed for the transpiration water loss experiment Line 247: Correct the sentence Line 282: Change to "exon"Section 4.5: The methodology for measurement of transpirational water loss is not clearly given. -Provide the details of the replicates (both biological and technical replicates) used in this study.
-Authors should consider measuring Ca2+ levels in the wild type and the mutant plants to establish a strong link between DORN1 and Ca2+ mediated drought tolerance
Author Response
Response to Reviewer 1 Comments
Dear reviewer,
We would like to thank you for providing helpful suggestions and comments. We have responded to your comments as written below.
We hope that you find our revision satisfactory.
In the manuscript entitled DORN1 is involved in drought stress tolerance through a Ca2+-dependent pathway, Wang et al have characterized the function of the DORN1 gene from Arabidopsis. Through various physiological and phenotypic analyses, the authors have concluded that DORN1 functions in drought tolerance. Compared to the wild type, the CRISPR/Cas9 edited plants of DORN1 showed decreased performance under drought conditions which was observed by increased water loss, stomatal opening, and reduced survival rate in the mutant plants. Overall, the manuscript is well presented with the required results supporting the conclusion.
Comments: Line 79: Change to "exon"
Line 98: Should be "dorn1 mutant plants"
Result 2.1: Only keep the results here. Delete the methodology part
Result 2.2 Describe how the survival rate was calculated.
Figure 5: Statistical analysis should be performed for the transpiration water loss experiment
Line 247: Correct the sentence Line 282: Change to "exon"
Section 4.5: The methodology for measurement of transpirational water loss is not clearly given. -Provide the details of the replicates (both biological and technical replicates) used in this study.
-Authors should consider measuring Ca2+ levels in the wild type and the mutant plants to establish a strong link between DORN1 and Ca2+ mediated drought tolerance
Responses to your comments;
- Line 79: Change to "exon"
[Author Response] Thank you for the suggestion. We have changed it to exon.
- Line 98: Should be "dorn1 mutant plants"
[Author Response] Thank you for pointing out our mistakes and giving the detailed suggestions. We have made modifications according to your suggestions. We gratefully appreciate for your valuable suggestion.
- Result 2.1: Only keep the results here. Delete the methodology part
[Author Response] Thank you for the suggestion. We have deleted the methodology part according to your suggestions.
- Figure 5: Statistical analysis should be performed for the transpiration water loss experiment
[Author Response] Thank you for the suggestion. We have made statistical analysis according to your suggestion.
- Line 247: Correct the sentence
[Author Response] Thank you for pointing out it, we have made modifications
- Line 282: Change to "exon"
[Author Response] Thank you for the suggestion. We have changed it to exon.
- Section 4.5: The methodology for measurement of transpirational water loss is not clearly given. -Provide the details of the replicates (both biological and technical replicates) used in this study.
[Author Response] We appreciate it very much for pointing our shortcomings, and we have added related description according to your suggestions. You can find it in 4.5.
- -Authors should consider measuring Ca2+ levels in the wild type and the mutant plants to establish a strong link between DORN1 and Ca2+ mediated drought tolerance
[Author Response] We appreciate it very much for this good suggestion, it has been reported in previous studies that Ca2 + levels in mutant dorn1. We have added this finding in the discussion section of the new revision.
Choi, Jeongmin, et al. "Extracellular ATP, a danger signal, is recognized by DORN1 in Arabidopsis." Biochemical Journal 463.3 (2014): 429-437.
Choi, Jeongmin, et al. "Identification of a plant receptor for extracellular ATP." Science 343.6168 (2014): 290-294.
Thank you for your careful review. We really appreciate your efforts in reviewing our manuscript. Your careful review has helped to make our study clearer and more comprehensive.
Reviewer 2 Report
I have revised the manuscript, and here are my comments: -
The authors need to check the manuscript's language, and the methods and discussion section should enhance.
Abstract needs to be reformulated regarding adding relevant results, and logistical and structural errors
Rephrase lines 40-61 for clarity
Clarify the objectives of the study at the end of the abstract and introduction
Check language by expert
Check and update the outputs of all references
Check the name of all devices, chemicals, and their companies
Check the abbreviations in the whole manuscript
Check the references style
Avoid old citations in the manuscript; note references 27 to 37
Author Response
Response to Reviewer 2 Comments
Dear reviewer,
We would like to thank you for providing helpful suggestions and comments. We have responded to your comments as written below.
We hope that you find our revision satisfactory.
- The authors need to check the manuscript's language, and the methods and discussion section should enhance.
[Author Response] Thank you for your suggestion, language presentation was improved with assistance from a native English speaker with appropriate research background.
- Abstract needs to be reformulated regarding adding relevant results, and logistical and structural errors
[Author Response] Thank you for underlining this deficiency. This section was revised and modified according to your suggestion.
- Rephrase lines 40-61 for clarity
[Author Response] Thank you for your suggestion, we have rewrited this part in the revised manuscript.
- Clarify the objectives of the study at the end of the abstract and introduction
[Author Response] Thank you for your suggestion, we have modified this expression throughout the text according to the comment.
- Check language by expert
[Author Response] Thank you for your suggestion, The manuscript has been thoroughly revised and rewritten by a native English speaker. we have carefully revised the language issue again.
- Check and update the outputs of all references
[Author Response] Thank you so much for your careful check, we modified it.
- Check the name of all devices, chemicals, and their companies
[Author Response] Thank you for pointing out our mistakes and giving the detailed suggestions. We have made modifications according to your suggestions. We gratefully appreciate for your valuable suggestion.
- Check the abbreviations in the whole manuscript
[Author Response] Thank you so much for your careful check, we modified it.
- Check the references style
[Author Response] Thank you so much for your careful check, we modified it.
Thank you for your careful review. We really appreciate your efforts in reviewing our manuscript. Your careful review has helped to make our study clearer and more comprehensive.
Round 2
Reviewer 2 Report
The authors have carefully processed all comments. The quality of the manuscript has increased significantly. I have no further comments.